# A Method for Heterogeneous Spatio-Temporal Data Integration in Support of Marine Aquaculture Site Selection

**Kate Beard [1],\*, Melissa Kimble [1], Jing Yuan [1], Keith S. Evans [2], Wei Liu [2], Damian Brady [2] and Stephen Moore [2]**

1   School of Computing and Information Science, University of Maine, Orono, ME 04469, USA;
    melissa.kimble@maine.edu (M.K.); jing.yuan@maine.edu (J.Y.)
2   School of Marine Science, University of Maine, Orono, ME 04469, USA; keith.evans@maine.edu (K.S.E.);
    wei.liu@maine.edu (W.L.); damian.brady@maine.edu (D.B.); stephen.moore@maine.edu (S.M.)
\*   Correspondence: kate.beard@maine.edu; Tel.: +1-207-581-2147

**Abstract:** Aquaculture site selection, like most site suitability analyses, requires the assembly and combination of multiple variables. Geographic information systems GIS and multi-criteria evaluation (MCE) based approaches are commonly used for aquaculture site selection and demonstrate the integration of various information sources relevant for siting aquaculture. These analyses, however, tend to be one-time and result in a fixed site suitability plan. Within a dynamic marine environment experiencing potential regime shifts, a siting support tool that integrates new and evolving spatio-temporal data has benefits. This paper presents a flexible Voronoi cell-based GIS model for marine aquaculture siting. Rather than a one-time specification of suitable locations, the approach uses similarity measures on the characteristics of Voronoi cells to find cells with similar characteristics. We calculate a weighted aquaculture site tenure value for Voronoi cells that have been or are occupied by aquaculture farm sites. High scoring cells suggest suitable sites and serve as targets for similarity queries. We apply the approach to a case study on the coast of Maine using an R Shiny application to demonstrate the use of the framework for finding sites with similar characteristics.

**Keywords:** aquaculture; site selection; GIS; spatio-temporal data integration; Voronoi cell-based

## 1. Introduction

Aquaculture production is expected to increase globally in response to expanding populations, growing demand for fish protein [1], and decreasing wild-catch fisheries [2]. The United States (US) enjoys a unique position for growth in its marine production. A recent Food and Agriculture Organization (FAO) study noted it among the top countries with the potential for profitably expanding marine aquaculture production [3]. Conflicts across limited coastal space may serve as a limiting factor to this growth potential. Identifying available areas that are suitable has thus become a critical concern for supporting and expanding aquaculture [4]. Indeed, the importance of the site selection process has been widely recognized in recent reports and research publications [5,6].

Numerous studies have used geographic information systems (GIS) and multi-criteria evaluation (MCE) for aquaculture siting and demonstrate the integration of many different data layers covering the spectrum from physical to economic to social factors [6–13]. The majority of these studies approach siting as a one-time analysis that specifies fixed areas as suitable for aquaculture. In a setting in which the environment and availability of data are changing, a flexible approach to siting that can take advantage of new data and changing conditions seems appropriate. This paper presents a GIS-based framework designed to support the integration of current data collection efforts and to accommodate

new data sources as they become available. We do not use the framework to specify site suitability directly. Rather, our approach allows users to explore sites with similar characteristics and we use existing farm sites to drive this exploration. We ground our approach on the idea that cells that are currently occupied by farm sites, especially over an extended period of occupancy, indicate sites with suitable physical characteristics. We calculate a measure of time-weighted farm occupancy for cells to suggest suitable site characteristics and allow interactive exploration for cells sharing similar physical characteristics.

This approach overcomes four key limitations of previous approaches. First, one-time analyses and fixed site suitability plans may not comfortably accommodate changing conditions. Second, one-off analyses are limited to currently available data without a pathway to incrementally refine analyses as new data become available. Third, input data is often restricted to static spatial layers. Finally, MCE requires the reclassification of all input variables to a common scale and assignment of weights to indicate the relative importance of different factors.

GIS-based models that support the integration of multiple geospatial data sets have become go-to tools for the analysis and identification of suitable site conditions. Nath et al. [7] were early advocates of GIS analysis for aquaculture siting and the use of GIS for aquaculture siting has continued to grow [6,8,10–15]. These studies demonstrate the combination of a range of data sets reflecting the requirements of different species, culture systems, and local context compiled as spatial layers. Most apply multi-criteria evaluation (MCE) techniques to connect individual criteria using additive or multiplicative models—with or without weights to indicate the importance of factors and obtain a suitability index or score for sites. Results typically take the form of suitability maps that designate the spatial distribution of areas with different levels of suitability for aquaculture production. In the next sections, we review some recent aquaculture siting studies that focus on shellfish aquaculture to document the state of current approaches.

A study for hanging culture of scallops in Funka Bay, Japan [8] used GIS and MCE with eleven criteria organized into three sub-models (biophysical, social–infrastructural, and constraints). The study required reclassifying these criteria to suitability scores ranging from 1 to 8 (most suitable), multiplied by a weight and summed for a final score. The authors noted the limits of their study in that several environmental parameters that influence scallop growth and survival such as dissolved oxygen, salinity, pH, wave height, water movement (tidal flow), fouling/disease/predators, pollution, and access to seed, were not available. A similar study for scallop aquaculture [16] included climate event indices; a monsoon index (MOI), and Oceanic Niño Index (ONI) integrated into the aquaculture site-selection model (SASSM) to assess climate variations on suitability. The authors noted that study limitations included missing variables; salinity, dissolved oxygen, and freshwater discharge, and constraints of satellite remote-sensing data, which only account for changes in surface waters. They suggested incorporating numerical model data to get temperatures with depth to better correspond with the hanging culture of scallops.

Two off-shore studies [9,11] document the use of GIS and MCE for oyster and mussel cultivation. Longdill et al. [9] demonstrated the use of different satellite imagery that included Advanced Very High-Resolution Radiometer (AVHRR) (1 km) sea surface temperature (SST) data, and Chlorophyll-*a* data from the Sea-viewing Wide Field-of-view Sensor (SeaWiFS) at 4 km and 1 km resolutions, noting that these represented the best available spatial and temporal data sets at the time. They used Parameter-Specific Suitability Functions (PSSFs) defined for each variable to convert the raw data to aquaculture suitability scores. They noted the advantage of the PSSF method over binary suitability scoring of 0 or 1 but also the level of subjectivity involved. Brigolin et al. [11] combined multiple spatial layers including MODIS satellite sea surface temperature and Chlorophyll-*a* concentration, current velocity from the NEMO ocean model, wave height from the SWAN model, and bathymetry. They used an MCE approach in which they normalized criteria, assigned a weight to each one, then aggregated the criteria to obtain a Suitability Index (SI) score. The SI scores were then assigned to 5 suitability

classes: 0–0.25, very low suitability; 0.25–0.35, low suitability; 0.35–0.50, medium suitability; 0.50–0.75, high suitability; >0.75, very high suitability.

A recent study [13] used GIS and MCE for bivalve marine aquaculture siting in the Baía Sul, Florianópolis, Santa Catarina State, Brazil. They aimed to overcome some previous modeling limitations identified as (1) arbitrary representation of aquaculture sites with well-defined boundaries; (2) modeling of spatial factors based exclusively on reclassification procedures; (3) lack of inclusion of different perspectives among stakeholders. The study identified 25 variables with input from several stakeholders which were converted to continuous maps to characterize the study area. While nominally continuous, several spatial layers were interpolated from point observations and thus subject to interpolation inaccuracies. The authors used the Analytical Hierarchy Process (AHP) [17] to generate weights for each variable which were then summed in an MCE analysis.

Several studies demonstrated the combination of GIS and MCE based approaches with various models (hydrodynamic, growth, biodepositional) [10,11,18]. Silva et al. [10] used a three-stage approach that included regulatory and social constraints, an MCE approach to determine suitability followed by a detailed site analysis using the FARM model [19]. Their study recognized temporal variability by interpolating maps of seasonal means and excluding areas that appeared as seasonally unsuitable. Final site suitability was obtained by combining legal and social constraints with MCE to generate suitability scores. The FARM model was applied to selected areas to assess potential production, socio-economic profits, and negative and positive environmental externalities. They noted limitations in their final suitability map due to subjectivity in assigning suitability ranges for factors. Newell et el. [18] used ShellGIS with ShellSIM [20], the shellfish growth model connected with a hydrodynamic flow model to allow the specification of a culture system and a specific species. This study had a dynamic component based on the hydrodynamic model with a focus on production capacity (stocking density that allows the sustainable harvest of shellfish to be maximized) but less consideration for broader siting criteria. Brigolin et al. [11] used the integration of MCE results with growth and depositional models to examine different scenarios and generate associated suitability maps rather than a single map solution.

These studies all noted data limitations and challenges in combining data of different spatial and temporal resolutions. They also all used some form of MCE with slightly different strategies to reduce the subjectivity in setting thresholds and assigning weights. The other similarity among these studies is that the results were fixed suitability maps.

With these challenges in mind, we aimed for an approach that could overcome some of them. A study [6], recognizing the issues with MCE, proposed an approach based on concepts from species distribution modeling (SDM). Their study used the locations of pangasius farms in the Mekong Delta in Vietnam as indicators of suitable sites in two different species distribution models; Mahalanobis Typicality and Maxent. The results of the SDMs do not imply suitability directly but rather indicate the similarity of locations to input farm sites. This approach has the advantage of avoiding subjective weight assignments and the need to normalize and reclassify multiple variables.

Our approach builds on the use of current aquaculture farm information as indicators, and in a sense, validation, of suitable conditions. In support of this strategy, we note that [11] found an alignment of current mussel farms with the zones they characterized as highly suitable. We incorporate the length of tenure of an aquaculture farm as a supporting indicator of site suitability.

Further objectives of our study were to accommodate diverse spatio-temporal datasets and development of an evolving and richer database over time; and shift from explicit, one-time mapping of site suitability to interactive map exploration of suitability based on similarity measures.

## 2. Materials and Methods

Maine has an active and growing marine aquaculture industry. The first official aquaculture lease, a mussel farm, was approved in 1973. Nearly a decade later, the first finfish aquaculture lease was approved, and finfish operations dominated Maine's aquaculture production for much of its early history. Recently, however, the shellfish sector (mainly eastern oyster, *Crassostrea virginica*) and marine

algae are showing rapid increases. Coastal Maine now has over 800 active aquaculture leases and short-term licenses for various shellfish and marine algae species. Challenges for siting aquaculture in Maine are the growing diversity in cultured species and costly data collection and analysis for a long and complex coastline (estimated at around 5600 to 8000 kilometersincluding offshore islands).

Under Maine's current regulatory structure, aquaculture siting decisions are made on a case by case basis by the state's Department of Marine Resources (DMR). Maine has two aquaculture lease types; Standard and Experimental, and a Limited Purpose Aquaculture license (LPA), that provide legal rights and protections to grow marine species in coastal waters. Each type specifies different property rights to its holder and indicates which marine species can be grown, and the duration and the renewability of the lease/license (see Table 1). The LPA licenses are intended to support low-cost test operations that may then be converted to standard leases.

**Table 1.** Maine aquaculture lease and license characteristics [21,22].

| Lease/License Type | Size Limit | Duration | Renewal | Notice Distance | Scoping Session | Public Hearing |
|---|---|---|---|---|---|---|
| Standard Lease | ≤4 km$^2$ | 20 years | Yes | 304 m | Yes | Yes |
| Experimental Lease | 016 km$^2$ | 1–3 years | No [d] | 304 m [a] | Maybe [b] | Maybe [c] |
| Limited Purpose License | ≤37 m$^2$ | 1 year | Yes | 91 m | No | No |

[a] The Maine DMR notifies the municipality, state, and federal agencies, shorefront property owners within 1000 m of the proposed site, and other interested parties at least 30 days prior to the public hearing. [b] Scoping sessions are at the discretion of the Maine DMR. [c] Yes if five or more comments are raised during the public comments period or the Maine DMR requests a public hearing. [d] Renewable if the experimental lease is designed for research purposes. Information from Maine DMR [21] and Maine Revised Statutes Annotated 12 [22].

The application process is decentralized with the initial siting choice proposed by the applicant. An application requires consideration of environmental concerns, other marine uses, and community reactions as obtained through scoping sessions and public hearings. The final decision rests with the Maine DMR Commissioner, based on a set of objective legal criteria. Specifically, a lease may not "unreasonably interfere" with riparian owners' land access, navigation, fishing, or other uses, support of ecologically significant flora and fauna, or public use or enjoyment within 304 m of government managed or conserved beaches, parks, docks, and land, and cannot have an "unreasonable impact" due to noise or light [21,22].

Marine aquaculture applications require that DMR scientists visit proposed site locations to verify site conditions. All standard and some experimental lease applications also require DMR scientists to use video to document the bottom environment, including plants and animals, and summarize the information in a report. Further, these applications also require that the DMR notify the municipality, state and federal agencies, shorefront property owners within 304 m of the proposed site, and other interested parties regarding the public comment period, the public hearing (a town-hall-style meeting), and opportunities to intervene.

Protection of benthic marine flora and fauna, especially protected and threatened species, is a socio-ecological constraint that has governed historical and recent siting of aquaculture in Maine. The U.S. Army Corps of Engineers requires that all Limited Purpose Aquaculture licenses and lease applications be sited outside delineated eelgrass zones and requires that potential loss of any other benthic vegetation be declared [22]. As a result, most existing aquaculture in Maine is sited above mud or sandy substrate.

Figure 1 illustrates the distribution of aquaculture lease and license (LPA) sites along the Maine coast as of 2019. Finfish aquaculture is primarily limited to the northern third of the state. The mid-coast region is dominated by shellfish aquaculture, with seaweed aquaculture recently beginning to show distribution over the full coastal range. For this study, we focus on the midcoast region which is the most active.

Comprehensive, uniform data coverage with a high level of detail is not easy to achieve for Maine's long coastline. Partial coverage of the coast however does exist in the form of different

data sets, collected by different groups, for different purposes. Our strategy was to develop a data integration framework that could accommodate these different data sets and that could fill gapsas new data become available with time. The integration framework we developed partitions coastal estuaries into what we call "characterization zones" based on a set of current point-based data observation stations. Different spatio-temporal datasets (time series) collected by various agencies or groups are summarized for each of the characterization zones. The history of aquaculture farm occupation further characterizes each zone Figure 2 diagrams this approach.

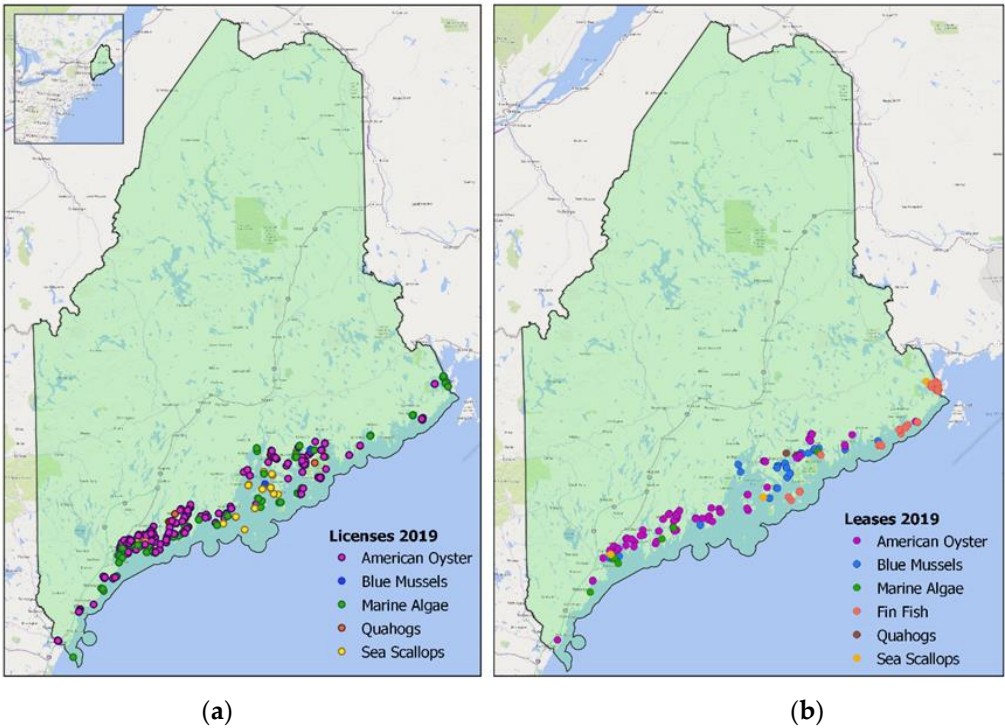

(**a**)　　　　　　　　　　　　　　　　　(**b**)

**Figure 1.** (**a**) The distribution of aquaculture license (LPA) and (**b**) distribution of lease sites along the Maine coast as of 2019. Basemaps from Bing maps © 2020 Microsoft.

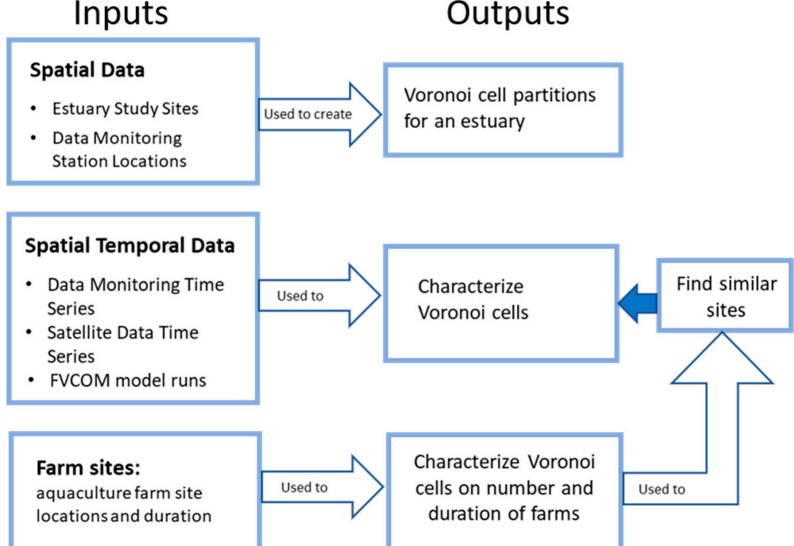

**Figure 2.** Overview of integration framework. We partition coastal estuaries into Voronoi cells and summarize spatial and temporal data sets for the Voronoi cells. Existing aquaculture site information is also summarized for each Voronoi cell. Cells with aquaculture sites serve as targets for finding cells sharing similar physical characteristics.

We demonstrate the framework for a set of currently available spatial-temporal data collections (summarized in Table 2). The Department of Marine resources collects water quality samples routinely along the coast of Maine at over 2800 fixed locations. This data collection effort started in 1990, continues to the present and into the foreseeable future. Each station is observed at least once a month during favorable weather months (Mar-Nov). Standard 100 mL samples are collected at stations located a few meters from shore with a field observation taken on water temperature followed by lab-based measurements for salinity and a fecal coliform score. These stations and their observation record form one of the longest and most consistent data collection efforts along the coast.

**Table 2.** Data sets used for characterizing Voronoi cells.

| Source | Type | Variables | Temporal Coverage | Temporal Frequency | Spatial Resolution | Spatial Coverage |
|---|---|---|---|---|---|---|
| DMR water quality samples | Point observations | Salinity, temperature, fecal coliform | 1990-present | monthly | point | Points spaced along the entire coast |
| LOBO buoys | Point observations | Salinity, Temperature, colored dissolved organic matter (CDOM), Chlorophyll a, dissolved oxygen (DO), nitrate photosynthetically active radiation (PAR), current speed and direction, turbidity, pH | 2013–2019 | hourly | point | 6 study sites |
| Ocean buoys | Point observations | Salinity, Temperature, colored dissolved organic matter (CDOM) Chlorophyll a, current speed and direction, turbidity, wind speed, wind direction, wave height | 2013–2019 | 20 min | point | 6 study sites |
| Landsat 8 | Satellite imagery | Sea surface temperature, Chlorophylla, turbidity | 2013–present | 16-day pass, cloud cover dependent | 30 m | Coast wide |
| FVCOM | Hydrodynamic model | Sea level, current speed, salinity, temperature | 2014 | Hourly | Varying (10–300 m) | Coast wide |

The Sustainable Ecological Aquaculture Network (SEANET), a 5 year, $20 million NSF funded project, identified six study sites distributed along the Maine coast. For each of these study sites, data have been collected or generated through deployed buoys, satellite imagery, and ocean circulation models. Land Ocean Biogeochemical Observatory (LOBO) buoys were deployed in six study sites for at least one growing season. The LOBO buoys sample 13 variables on an hourly basis. As part of SEANET, additional ocean monitoring buoys were deployed on the offshore perimeter of estuaries. These monitoring buoys sample on a twenty-minute interval on 14 variables that include temperature and salinity among others (see Table 2). These deployed buoy locations cover the period from 2013 to 2019. Landsat-8 Operational Land Imager (OLI) and the Thermal Infrared Sensor (TIRS) provide an enhanced data source with a relatively high spatial resolution for coastal regions. Under SEANET, Landsat 8 satellite imagery was assembled and processed to derive sea surface temperature (SST), Chl *a*, and turbidity [23]. This imagery data set covers from 2013 onward.

Another SEANET effort supported the development of three-dimensional hydrodynamic models for the Maine coast. These models are based on unstructured-grid Finite Volume Coastal Ocean Model (FVCOM), which has the advantage of accurately following complicated coastlines by using unstructured triangle elements [24–27]. The model domain covers a wide shelf area in mid-coast Maine and major estuaries including the Kennebec River, Androscoggin River, Sheepscott River, Damariscotta River, Medomak River, and St. George River. The unstructured mesh allows a large model domain in the estuaries with spatial resolution as high as 10 m within the estuaries. The model has simulated the entire year of 2014. Outputs include two-dimensional hourly data for sea level and three-dimensional hourly data for temperature, salinity, and current velocity. Model outputs have been validated using observational data sources that included the LOBO buoy time series data.

The Department of Marine Resources has partitioned the coast of Maine into 45 growing areas for the management of shellfish harvesting (See Figure 3). These growing areas generally correspond to individual estuaries or bays. As these have official governmental standing, these form our top-level partition and unit of analysis.

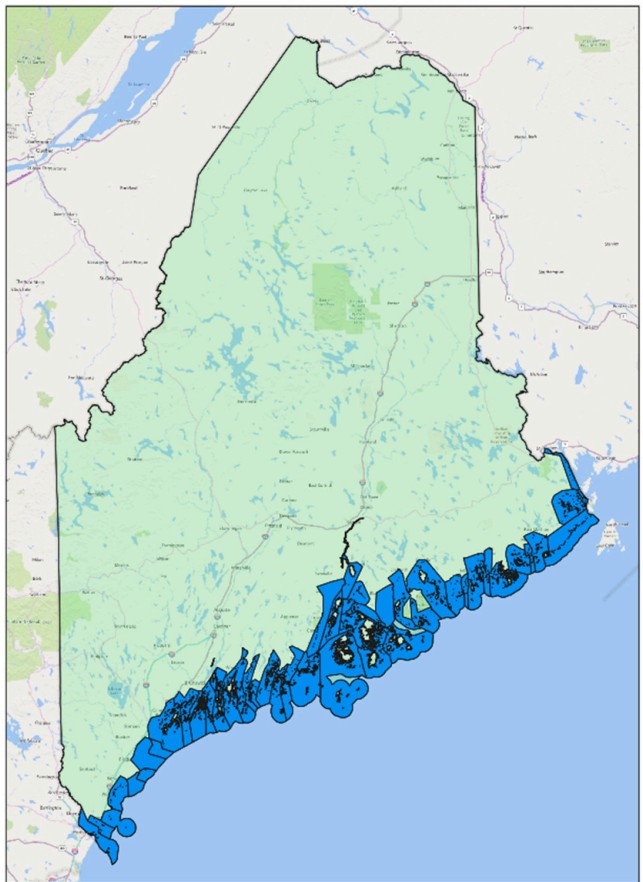

**Figure 3.** Shellfish Growing Areas as defined by the Maine State Department of Marine Resources. Basemap: Bing maps © 2020 Microsoft.

We next partition the bays or estuaries within each of the Growing Areas into Voronoi cells [28] using the fixed locations of monitoring stations (DMR, LOBO, Ocean Monitoring stations). These Voronoi cells form our characterization zones. The defining characteristic of Voronoi cells is that they are the areas closer to a given point than any other point in a generating set (see Figure 4). Voronoi cells thus represent a proximity zone for each data monitoring station. Other important attributes of Voronoi cells are that they are computationally easy to generate, and they can be updated locally. If a new observation station is added, the Voronoi partition can be updated locally without having to recompute the entire partition [29]. Given Maine's convoluted coastline, we had to modify standard Voronoi cell construction based on Euclidean distance to one based on the shortest path distances within the water. We used the Cost Allocation function within ESRI ArcGIS Pro (version 2.3) with an input cost layer that assigned a minimal cost to water areas (e.g., 1) and a high cost to land areas (e.g., 500) which effectively restricted cost distance paths to water areas.

The Voronoi cells become the characterization zones for various physical variables pertinent to aquaculture siting. For each Voronoi characterization zone, we summarize the available time-series data as quantiles. The point-based data observation stations which define the Voronoi zones contribute their associated times series towards characterizing their respective zones. For spatially extensive data types such as satellite imagery raster layers or hydrodynamic models results, we aggregated values

to the Voronoi characterization zones. Landsat 8 derived raster layers for sea surface temperature (SST), Chl *a*, and turbidity were overlaid with the Voronoi zones and quantiles were computed for each variable and each zone. Similarly, FVCOM model generated layers were overlaid with the Voronoi zones and quantiles were generated for each variable and each zone. Figures 5 and 6, respectively, illustrate examples of Landsat derived sea surface temperature and FVCOM salinity data overlain with the Voronoi zones. The values of Landsat pixels or FVCOM grid points falling within each Voronoi cell are summarized and stored as quantiles.

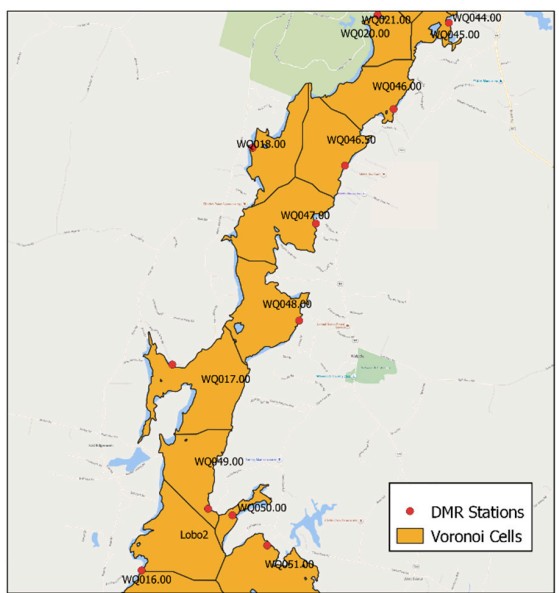

**Figure 4.** Voronoi cells generated from DMR monitoring stations plus LOBO buoy locations in the Damariscotta River Estuary. The labels are DMR station numbers represented by a concatenation of the Growing Area label and a station number. Each observation point captures water areas closer to it than any other point. Base map by Bing maps © 2020 Microsoft.

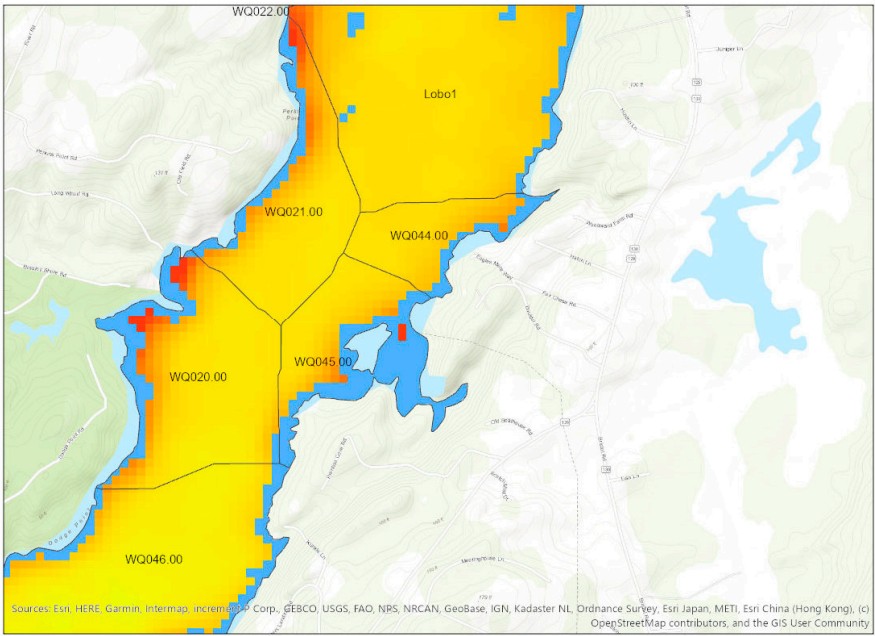

**Figure 5.** Landsat 8 derived August mean sea surface temperature (SST) data associated with Voronoi cells in the Damariscotta River. The values of SST pixels falling within a Voronoi cell are summarized as monthly or seasonal quantiles.

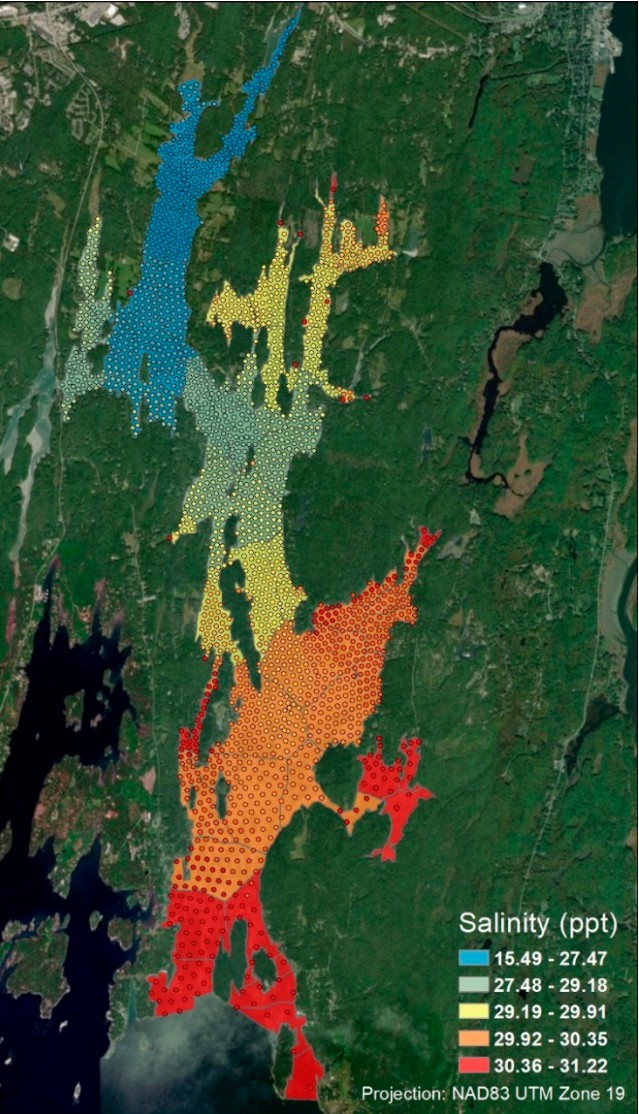

**Figure 6.** FVCOM model results summarized as a monthly mean (August mean salinity) associated with Voronoi cells in New Meadows River Estuary. All modeled values within a Voronoi cell are summarized as quantiles. Imagery base map sources: Esri, DigitalGlobe, Earthstar Geographics, CNES/Airbus DS, GeoEye, USDA FSA, USGS, Aerogrid, IGN, IGP, and the GIS User Community.

We note that quantiles can be generated for different temporal granularities (e.g., months, seasons) depending on a desired or appropriate temporal resolution. For this case study, we demonstrate monthly-based quantiles. Given the current data sets, a Voronoi cell can have up to 12 monthly DMR salinity, temperature, and fecal coliform score quantile vectors, 12 monthly Landsat SST, Chl *a*, and turbidity quantile vectors and 12 monthly FVCOM temperature, salinity, and current velocity quantile vectors.

For each Voronoi characterization zone, we further calculate a normalized aquaculture farm tenure value. Normalized tenure for a cell is calculated as a function of the number of marine aquaculture farms and their duration in the cell. The formula for normalized cell tenure (NCT) is

$$NCT_i = \frac{\frac{\sum_j T_j}{n_i}}{A_i} \tag{1}$$

where $T_j$ is the tenure for lease $j$, $n_i$ is a number of farms in cell $i$, and $A_i$ is the area of cell $i$. This function gives greater weight to longer tenure leases or persistent license renewals with the reasoning that lease/license persistence at a site may indicate higher suitability. As an example, a cell with five one-year leases would have an NCT numerator of 1 whereas a cell with a single five-year lease would have a value of 5. Figure 7 illustrates NCT values for the New Meadows and Damariscotta River estuaries.

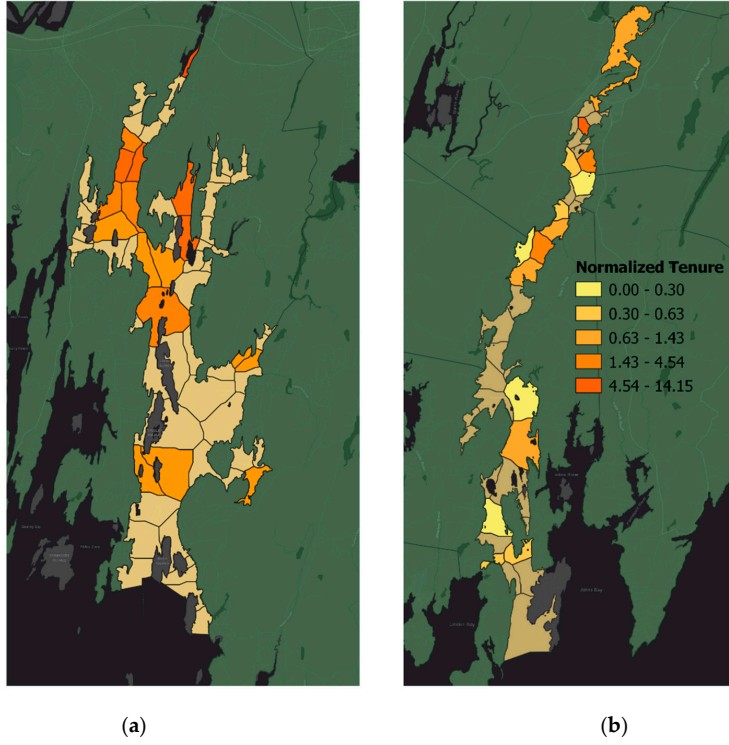

**Figure 7.** (**a**) Choropleth map of NCT values for New Meadows River estuary and (**b**) Choropleth map of NCT values for Damariscotta River estuary. Basemap © OpenStreetMap contributors, and the GIS user community.

We used R Shiny [30] to implement this framework. Our R Shiny application displays the Voronoi partitions for an estuary and can display associated values as choropleth maps. The opening display shows NCT values for the Voronoi characterization zones that have been occupied by aquaculture farms.

Within the R Shiny application, the Voronoi characterization zones can be queried to identify cells with similar characteristics. This query functionality employs cosine similarity measures between the quantile vectors associated with each Voronoi characterization zone. For example, all Voronoi characterization zones containing DMR observed variable quantiles can be assessed for similarity on their monthly salinity, temperature, and fecal coliform quantiles.

We used species distribution models (SDM) (Maxent and Mahalanobis Typicality) as a basis to evaluate NCT value alignment with model predicted site suitability scores. SDMs are primarily used to predict species locations based on correlations with selected environmental variables. Falconer et al. [6] used SDMs creatively to extrapolate aquaculture site suitability using farm sites to represent species occurrence data. We ran SDMs on two midcoast estuaries with the most aquaculture farm sites using the R package dismo [31]. For each model, the current aquaculture farm site coordinates were used to represent species occurrence data (there is no consideration for the duration). Environmental predictor variables for the models were selected to take into account limiting factors for shellfish growth. These included the following quantile values associated with the Voronoi cells: upper quantile of temperature for summer months, lower quantile of temperature for winter months, the lower quantile of salinity for summer, and median salinity for winter. High concentrations of fecal coliform affect the quality of shellfish or marine algae. Fecal coliform scores can spike in summer months due to heavy rain

events thus we used the upper quantile of fecal coliform scores in summer. Landsat derived sea surface temperature [23] was also used with the same quantiles as DMR water temperature observation. The predictor variables were converted to raster layers to create a raster stack. We used a function in the dismo package [31] to sample random points as background data. We extracted values from the raster stack for occurrence points and random background points. These data were input to SDM models to predict aquaculture suitability. Models were evaluated based on Area Under the Receiver Operator Curve (AUROC or AUC score for short).

Using SDM predictions as measures of suitability [6], we regressed them against our time-weighted aquaculture farm site density (NCT) values. We used quantile regression to fit SDM and NCT values as higher densities of aquaculture sites are expected in regions with high suitability [32].

## 3. Results

### 3.1. R Shiny Application

The results of this work are a GIS-based framework that supports aquaculture lease or license site evaluation by prospective farmers. We implemented the framework as an R Shiny web application (see https://rshiny.spatialmsk.com/CosSim/). Instead of generating static suitability maps, the developed framework supports interactive queries on the Voronoi characterization zones. Given the characterization of a Voronoi cell, users can search for cells sharing similar profiles. As an example use of the framework, a user can select a Voronoi characterization zone with a high NCT value (indicating high farm occupancy and/or long tenure). The application returns a list of Voronoi characterization zones ranked on their similarity to the selected cell.

For any Voronoi cell selected by a user, the application reports a similarity to all other cells. Users are typically not interested in similarity scores to all zones, just the top few. The application thus accommodates fast interactive filtering of the similarity scores from most to least similar through a slider widget. Through the slider, the user can set a similarity threshold and cells with values above the threshold are highlighted on the map. Similarity scores can also be viewed in a table in rank order. Instead of using high NCT scoring cells as targets, users can target specific species. With current farm sites symbolized on the map by species type, a user can, for example, select a cell occupied by mussel farms and find cells most similar in their environmental profile to the test cell. Figure 8 illustrates a screenshot of the Shiny interface.

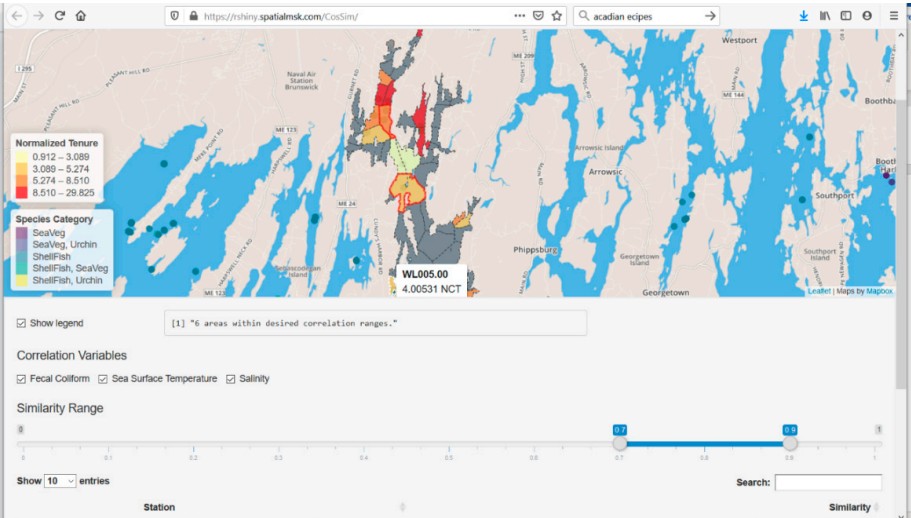

**Figure 8.** A screenshot of the R Shiny interface. When a user selects a Voronoi characterization zone, the application reports the most similar zones based on the underlying physical properties. The most similar cells are highlighted in red on the map.

### 3.2. SDM Modeling and NCT Comparison

The results of the SDM models are shown in Figure 9. The raw values of the models score the Voronoi characterization zones on similarity to cells containing farm sites. The results of the Mahalanobis distance model indicate the degree to which the values of environmental variables at a location are typical of conditions at a farm site. Similarly, the Maxent output can be interpreted as predicted probability of suitable conditions for the modeled species. High scores indicate the most similar sites and by extension the most suitable sites. We interpret these results in a qualitative sense.

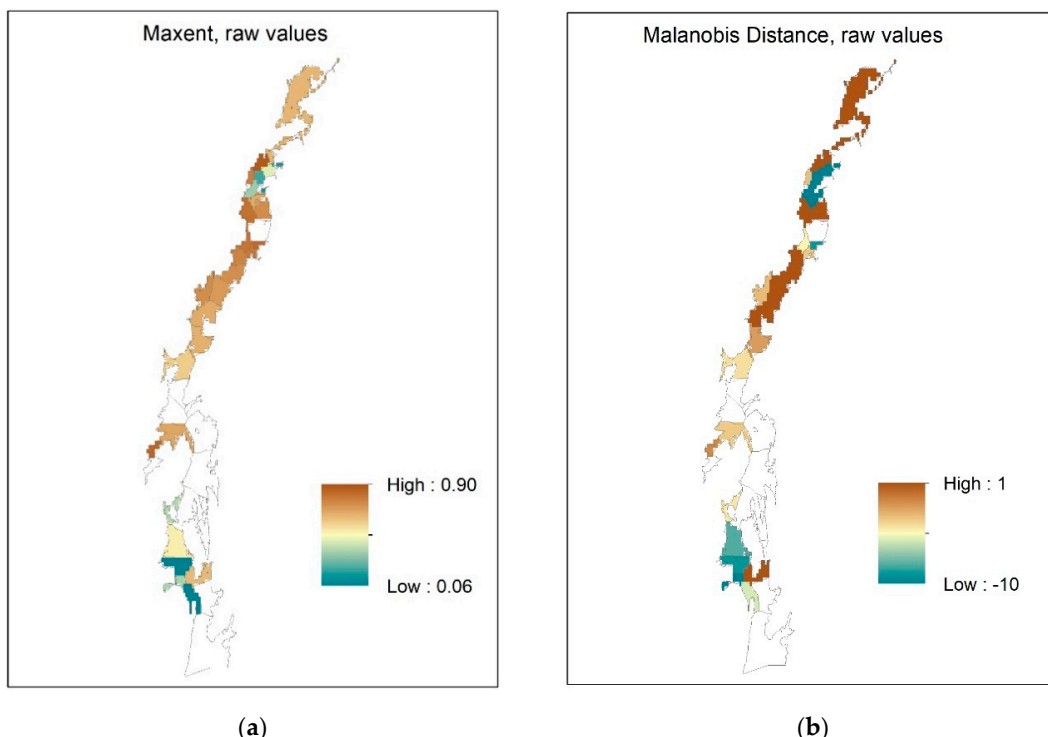

(**a**)                                   (**b**)

**Figure 9.** (**a**) Raw scores for the Maxent model at Damariscotta and (**b**) raw scores for Mahalanobis models at Damariscotta. Scores apply to Voronoi cells. The results show a general agreement on site similarity /suitability between the models.

Area under the receiver-operator curve (AUC) is a common measure of predictive accuracy for SDM. For the AUC values (see Table 3), the rule of thumb is 0.5–0.7 low, 0.7–0.9 moderate, >0.9 high. The Mahalanobis and Maxent models fall in the moderate range which is likely due to the small sample size used for this test. In terms of the max true positive rate (TPR)/sensitivity and true negative rate (TNR)/specificity, Mahalanobis performs best. The performance of the SDM models was not the most important outcome. Rather our objective was to use the model results to get general measures of suitability for comparison to our normalized cell tenure (NCT) values. The quantile regression of NCT and SDM scores (see Table 4) showed significant correlations which tended to be highest at the highest quantiles.

**Table 3.** Results from the species distribution modeling (SDM) assessment.

| Model | AUC | COR | max TPR+TNR |
|---|---|---|---|
| Mahalanobis Distance | 0.72 | 0.08 | 1.00 |
| Maxent | 0.69 | 0.27 | 0.74 |

**Table 4.** Results of the top three quantiles (70th, 80th, 90th) for the quantile regression of the normalized cell tenure (NCT) values on SDM model scores. Results of the quantile regression indicate that our NCT values are well correlated with the SDM modeled suitability scores.

| SDM | Quantile Regression Slope | | | | | |
|---|---|---|---|---|---|---|
| | 70th | Pr | 80th | Pr | 90th | Pr |
| MAXENT | 0.47 | 0.0006 | 0.58 | 0.000001 | 0.8 | 0.000001 |
| MD | 5.97 | 0.016 | 9.14 | 0.0015 | 11.28 | 0.00011 |

## 4. Discussion

This study investigated a new GIS-based framework for evaluating aquaculture site suitability. The approach uses Voronoi cells adapted to partition coastal estuaries into characterization zones. One objective of the developed framework was to support the integration of current data observation efforts along the coast as well as accommodate new data collection going forward. The Voronoi cells are built around current point-based observation stations. Our starting data collection points are the stations monitored by the Maine DMR. The SEANET LOBO observation sites were added to this set as they were deployed over the study period. There is thus flexibility within the framework to add observation points and increase the spatial resolution of the characterization zones over time. For the highly crenulated midcoast region of Maine, adjacent estuaries can exhibit quite different behaviors so having comprehensive observation coverage becomes critical relative to smoother coastlines. Comprehensive coverage of the coastline by the Maine DMR water quality observation stations made them a practical starting point. While Maine has a relatively dense observation network for constructing the proposed framework, any coastal region with a deployed monitoring network or considering one, could take advantage of this framework.

Our rationale for using Voronoi partitions was based on the diversity of data types to be integrated and because they are proximity-based and tend to be compact. To combine any geospatial data sets with different spatial resolutions, some common spatial unit must be adopted which typically requires some compromise on scale. Combining diverse resolution raster data generally involves converting all layers to the coarsest spatial resolution (pixel size). To integrate point-based observations then involves interpolating points to a raster representation as in [13]. Raster representations have the seeming advantage of finer spatial resolution and more comprehensive spatial coverage but have attendant disadvantages. They can be more subject to missing values and in the presence of high spatial autocorrelation where adjacent values are very similar, they impose a higher storage cost for little to no information gain. For this study, the Voronoi cell sizes ranged from 0.4 to 3.1 km$^2$ with a median value of 48 sq km. The degree of spatial autocorrelation in sea surface temperature (SST) based on 30 m pixels was very high (*Morans I* = 9). The estimated range (correlation distance) for a fitted variogram for SST was 4000 m, which closely matches the median cell dimension. There was thus little loss in spatial detail in employing Voronoi cells and a processing and storage advantage.

Many of the data sets we used had missing values in time. The Landsat 8 satellite imagery derived products which offer high-resolution spatial coverage, suffer from sporadic temporal coverage due to the 16-day repeat cycle and frequent cloud coverage. The DMR sampling data had less frequent coverage in winter months and the LOBO buoys were not deployed in winter months. By using the Voronoi cells to create time-based summaries of different variables from different sources we limited the impact of missing values. As the Voronoi cells are proximity-based, we might expect that quantiles on the same variable (but from different sources) assembled for the Voronoi characterization zones to be comparable and substitutable. Comparisons among quantiles for a subset of test cells showed a high overlap. The overlap of distributions of the same variables from different sources within a Voronoi cell was exploited to avoid missing variable quantiles and also provided a basis for quality and validation checks.

The Voronoi characterization zones provide a basis for integrating on-going time-series-based physical data. The variables employed in this case study represent examples of dynamic physical variables that change seasonally and may show interannual variation. For this case study, we summarized time series as monthly quantiles, but seasonal or other time-based summaries including time-based anomalies are possible within the framework.

The Voronoi characterization zones served to integrate diverse time-based physical data sets. As a GIS-based framework, the Shiny application can incorporate additional spatial data layers to indicate constraints that might apply. Within the application, layers could be added to indicate ecological constraints such as the presence of eel grassbeds, marine protected areas, or other use constraints such as navigation channels, moorings, or other fishing grounds.

Our approach shares similarities with [6] in that we use the spatial patterns in current aquaculture farm sites as implicit evidence of suitability with the addition of farm duration at a site. The set of past and current aquaculture farm sites indicate locations that have passed muster on governmental regulations and constraints and the addition of duration suggests some indication of production viability. We use mapped NCT scores to help users identify cells reflecting suitable physical characteristics. Instead of designating suitability explicitly, our application supports exploration for cells sharing similar physical characteristics to cells occupied by farm sites. The application currently uses cosine similarity based on monthly quantiles for temperature, salinity, Chl *a*, and current velocity.

Similarity-based queries have been applied in several application areas where specific selection criteria can be challenging to express. Examples include image and document search where users supply an example and are returned a ranked list in response. Specification of aquaculture site suitability shares similar difficulties in expression. MCE models require the reclassification of all input variables to some common scale (e.g., 1 "highly unsuitable," 2 "unsuitable," 3 "intermediate," 4 "suitable," and 5 "highly suitable"). For physical variables such as temperature and salinity, the specification of hard boundaries on suitability ranges tend to be subjective and vary with species. Fuzzy reclassification allows a continuous scale from 0 to 1 but can complicate interpretation. The use of similarity measures avoids the need to specify hard thresholds on continuous variables and the variable reclassification and weighting typically required in MCE models.

A similarity search relies on specifying a target. In our application, farm-site occupied cells perform this service. Our NCT scores capture the time dimension (how long leases or licenses have persisted in a cell) for display on a map. To evaluate NCT scores as representative of site suitability, we regressed them on SDM model results (a surrogate measure of suitability). Using quantile regression, the results indicate a high correlation in the upper quantiles suggesting high agreement with the highest SDM scores. With results indicating that site similarity can reasonably replicate suitability as demonstrated in [6], we opted to employ cosine similarity metrics to assess site similarity in place of the more complex SDM modeling.

A final advantage of the similarity-based approach is that it is not bound to particular species. Prior aquaculture siting studies assembled, reclassified, and weighted variables to be pertinent for specific species requirements. By displaying aquaculture farm sites by cultured species type on the map, users are supplied with search targets by species type.

Rather than supporting just a one-time analysis, we think that this application has the potential to continue to grow as a siting support tool. The open-ended framework that supports the addition of new data and new farm sites as they are approved suggests a trajectory in which data supporting the application can become stronger over time.

**Author Contributions:** Conceptualization, K.B.; M.K., and J.Y.; data curation; W.L.; formal analysis, K.B.; M.K., and J.Y.; investigation, K.B., M.K., J.Y., and K.S.E.; methodology, K.B., M.K. and J.Y; resources, W.L., D.B. and S.M.; software, M.K.; Validation, K.B. and J.Y.; writing—original draft, K.B.; J.Y.; K.S.E., and W.L.; writing—review and editing, K.B.; M.K.; K.S.E., and D.B. All authors have read and agree to the published version of the manuscript.

**Funding:** This research was supported by the National Science Foundation award #11A-1355457 to Maine EPSCoR at the University of Maine.

**Conflicts of Interest:** The authors declare no conflict of interest.

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
