# Peer review of "A Method for Heterogeneous Spatio-Temporal Data Integration in Support of Marine Aquaculture Site Selection"

_jmse, doi:10.3390/jmse8020096_

Round 1
Reviewer 1 Report
This is an interesting and innovative paper. It is also well written. Only miron change are required. I have a few suggestions for organization. First, the methods are too long and have too many results and figures. Much of this is actually results. The Results are too brief, especially the section on SDM which seems like it contains more methods about the range of values without clearly stating the results of the SDM NCT comparison. Also, unless someone really reads carefully, they might conclude that SDM is part of the approach rather than a type of calibration. More emphasis on SDM as a check would be useful.
Finally, I wonder why depth did not enter the example more prominently. Bathymetry is always known and always used in marine SDM. It could be in the NCT analysis as well.
Reviewer 2 Report
The article is a very interesting, based in a new approach to finding sites to aquaculture. Can be accept in the present form.
Please check line 55, I tink is a coma before "...were early ..."